# Joint Capsule Segmentation in Ultrasound Images of the Metacarpophalangeal Joint using Convolutional Neural Networks

**Nelson Martins**
Neadvance, Machine Vision, SA
Braga, Portugal
`nmartins@neadvance.com`

**Eva Costa**
Neadvance, Machine Vision, SA
Braga, Portugal
`ecosta@neadvance.com`

**Diana Veiga**
Neadvance, Machine Vision, SA
Braga, Portugal
`dveiga@neadvance.com`

**Manuel Ferreira**
Neadvance, Machine Vision, SA
Braga, Portugal
`mferreira@neadvance.com`

**Miguel Coimbra**
Instituto de Telecomunicações
FCUP,* Porto, Portugal
`mcoimbra@dcc.fc.up.pt`

## Abstract

This work addresses the automatic segmentation of the joint capsule in ultrasound images of the metacarpophalangeal joint using an adapted version of the well known $UNet$ model. These images are used in the diagnosis of rheumatic diseases, one of the main causes of impairment and pain in developed countries. The identification of the joint capsule gives important clues about the presence or Rheumatoid Arthritis. This structure can be used to extract metrics to help quantify the disease stage and progression. The solution proposed here has the potential to reduce the burden on the radiologists as well as the subjectivity of the diagnosis by providing quantitative measurements, such as the synovitis area. The proposed approach was compared with two other works present in the literature. Results show that our solution outperforms the two reference methods with 90% of the joint capsules identified with a $DICE$ higher than 0.67.

## 1 Introduction

Rheumatic disorders refer to a group of conditions that cause pain in the joints and connective tissue with a serious impact in the health of the patients as well as high social and economical implications Scheel et al. [2005]. In fact, they are one of the main causes of pain in the developed countries, with higher incidence in older people but also with a significant incidence in the young. Given its chronic nature, people with rheumatic problems need constant vigilance and medication so that the disease progression is slowed (Schueller-Weidekamm [2009]). This puts a pressure in the early detection, since it allows better treatment outcomes and, in the long run, reduce the degenerative effects of the disease. The problem is that the early diagnosis is not easy because of the subtlety of the symptoms. To help the rheumatologists in this task, several tools and guidelines are available (Aletaha et al.

---

*Faculdade de Ciências, Departamento de Ciências de Computadores, Universidade do Porto

1st Conference on Medical Imaging with Deep Learning (MIDL 2018), Amsterdam, The Netherlands.

[2010]). They are based on the visual inspection and patients testimony, laboratory tests and imaging tests. Regarding the imaging tests, they allow the visualization of the patient's internal anatomical structures, which can be used to identify pathology patterns, such as swelling, inflammation and bone erosion. From all imaging modalities, ultrasound has gained interest over the years, due to its advantages, such as low cost, acceptability by the patients and no ionizing radiation. The problem with this technique is the difficulty in reading the images (Meenagh et al. [2007]). Image processing tools can be used to help overcome this limitation, by automatically identify and quantify findings. Following this idea, we propose the use of Convolutional Neural Networks (CNN's) to automatically segment the joint capsule in ultrasound images of the metacarpophalangeal joint (MCPJ). In this way, we expect to reduce the subjectivity of the diagnosis and help in the detection and follow-up of Rheumatoid Arthritis (RA). From all the possible acquisition protocols, we decided to use the dorsal-longitudinal view since it is one of the most informative view to study the joint capsule, Fig. 1.

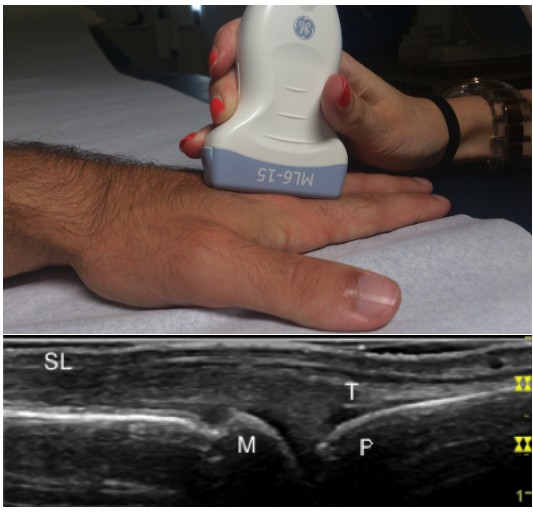

Figure 1: Acquisition apparatus of the ultrasound images of the longitudinal view of the MCPJ; SL - Skin line; M - Metacarpus; P - Phalange; T - Extensor tendon.

The joint capsule is not always visible, especially in healthy patients. However, when the inflammation of the joint capsule is present it becomes visible due to the accumulation of synovial fluid. This condition is known as synovitis and is classified in four grades, as shown in Fig. 2.

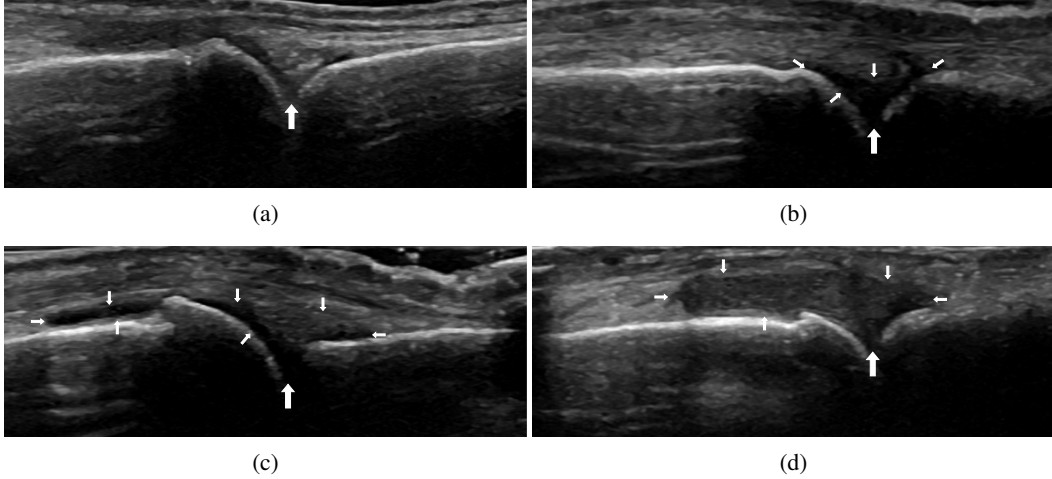

Figure 2: Example of ultrasound images of the second MCPJ with different joint capsules aspects and synovitis grades. Small arrows indicate the joint capsule extension and big arrow indicates the joint. a) Grade 0; b) Grade 1; c) Grade 2; d) Grade 3.

Fig. 2 shows the different synovitis appearances in each stage, being 0 the absence of inflammation and the remaining increasing stages of inflammation. From Fig. 2, it is also possible to verify that the joint capsule is a darker region that grows from the space between the head of the metacarpus and phalange and that its size is proportional to the synovitis grade.

The manuscript is organized as follows, in the section 2 a brief review of the state of the art is presented, followed by the automatic segmentation of the joint capsule in section 3. The results and discussion are presented in section 4, while section 5 is devoted to the conclusions and future work.

## 2    State of the Art

The detection of the joint capsule is still an open problem. Given the problem complexity, Veronese et al. [2013] proposed a semi-automatic method: the algorithm starts with a pre-processing step, which is based on the minimization of local variations and the assumption that the noise follows a Rice distribution. Then, the user must choose three points, two in the synovial limits in the bones and the third is used to define the synovial extension between the limits of the bones and articulation. After that, two sets of active contours are created. The first defines the contours of the hyper proliferation of the synovial membrane and is used to initialize the second contour that will separate the synovial membrane and the soft tissues. In Nurzynska and Smolka [2016a] and Nurzynska and Smolka [2016b], the authors proposed an automatic method that starts with the identification of the skin border and bones; then a coarse segmentation is performed using a global threshold obtained from the accumulative histogram (at 65%). This coarse segmentation is improved using the skin border, the bones' segmentation and a confidence map. The author reported satisfactory visual results from experts. In Martins et al. [2018] a split and merge approach with a refinement step was proposed. The algorithm uses the annotations of the bones and extensor tendon to create a region of interest, the SLIC algorithm is then used to split this region in small clusters. These clusters are then merged using a specific region growing with shape constraints. A $DICE$ higher than 0.7 in 60% of the images was reported. Regarding the use of Deep Learning (DL) techniques in the ultrasound images of the MCPJ, there were no works found in the literature. The closest one was the work of Golan et al. [2016], where CNN's were used to identify hip dysplasia. In Litjens et al. [2017], it is presented a broad overview of the existing methods using DL in the medical field. Regarding the segmentation of 2D images, the authors emphasize the $UNet$ model, published by Ronneberger et al. [2015], which lead to a significant improvement in the segmentation of cells in electron microscopic stacks using a small training set. Given that the database available in the present work is also limited, it was decided to use the $UNet$ as a starting point for this task.

## 3    Proposed Work

In this work, CNN's will be used to identify the joint capsule in ultrasound images of the MCPJ. CNN's are the current state of the art approach to solve several image processing problems. From classification to segmentation, they achieved better results than most of the previously used techniques Ronneberger et al. [2015], Nurzynska and Smolka [2016b], Russakovsky et al. [2015]. One problem is that the typical approaches using CNN's require an extensive database, to properly obtain the network parameters. From the state of the art revision, we conclude that the $UNet$, Ronneberger et al. [2015], could be a viable solution for this task. Next it will be presented in more detail.

### 3.1    Model adaptation

In short, the $UNet$ tries to capture local and global information in a single model, by adding skip connections before every $max - pool$ operation. This ensures that the local information lost after the $max - pool$ is kept and added later in the network. This model was formulated to solve multi-object segmentation problems with touching boundaries. In the present work, the problem is the detection of a single object (joint capsule) and consequently, the $UNet$ model needs to be adapted. Those changes will be addressed in the following subsections.

### 3.1.1    Architecture

The adapted architecture, proposed for this problem, is depicted in Fig. 3.

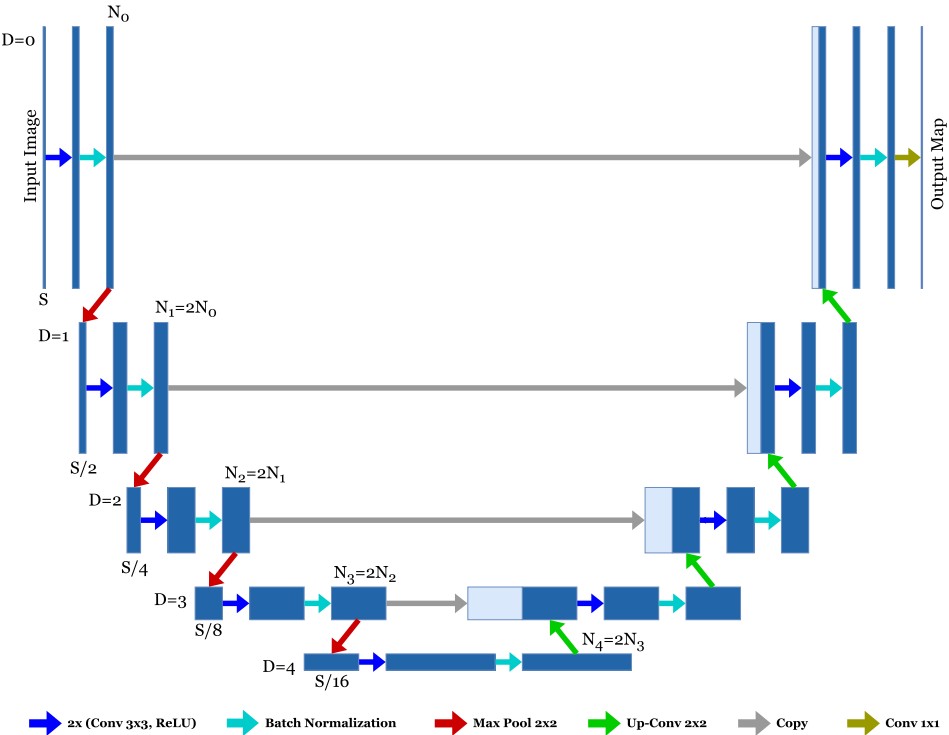

Figure 3: Proposed $UNet$ model, adapted from Ronneberger et al. [2015]. Normalizing layers were added before every $max - pool$ operation; $same$ padding in all $Conv$ layers, instead of $valid$. $N_x$ stands for the number of filters, $D$ for depth and $S$ for the spatial size of the feature map.

The first architectural change is the use of zero padding in the $Conv$ layers, instead of the valid values. The original model uses the valid values with a mirror padding pre-processing to accelerate the training, however this was not observed during preliminary tests and therefore, we decided to simplify the model and use zero padding in all $Conv$ layers. The second change is the use of batch normalization layers before the concatenation steps. This adds a regularization effect by ensuring that the concatenated feature maps have the same order of magnitude. In the last $Conv$ layer, the Softmax activation function was replaced by a Sigmoid function, since the joint capsule segmentation is a binary problem and it is not necessary to use a layer with multi-class properties.

### 3.1.2 Data Augmentation

Along with the $UNet$ architecture, Ronneberger et al. [2015] used a special data augmentation technique to compensate the reduced number of training samples. In that work, elastic transforms were used to simulate the possible deformations that a cell can experience. In the MCPJ images, this technique is not usable, since the deformation is constrained by many other factors, like the presence of bones, tendons and rotation points. Because of that, we decided to create a different set of augmentation techniques. These techniques are expected to improve the generalization capabilities of the model by creating additional artificial data with different characteristics, not present in the original database:

- Scale - simulates anatomical differences in size (bigger/smaller joints) and the ultrasound device acquisition parameters: depth and probe width;
- Horizontal shifts - simulates different probe positions along the joint;
- Illumination - simulates different grayscale maps used by the ultrasound device operators;
- Noise - simulates the noise from different equipment;
- Horizontal flips - simulates the transformation that occurs when the operator rotates the probe $180°$.

The input image was scaled using a bicubic interpolation, while the output mask was scaled using the nearest neighbor interpolation. The range of the scale parameter was empirically selected to be between 0.8 and 1.2. The horizontal shift was done using a random crop operation inside the valid zone of the resulting scaled image. The illumination was achieved using a random scalar value added to the input image between -20 and 20. The noise was added to the image using a random noise generator with mean zero and standard deviation 10. The horizontal flips were done with a simple mirroring operation in the image. In each epoch, 10 new images are randomly created for each original image. This way, it is expected to increase the variability of the training data.

### 3.1.3 Other changes

The input image was trimmed to 256 lines and 736 columns to avoid downsampling and upsampling rounding errors and to reduce the bottom dark region of the images. The optimization method was the adaptive momentum (Adam) with default parameters. The objective function was the mean squared error instead of the one proposed in Ronneberger et al. [2015], which took into account the multi-class segmentation with touching boundaries, which is not adequate for the present situation. Since each layer has its own number of neurons, it was decided to follow the rule that after every $max - pool$ operation the number of neurons is doubled. Thus, only the number of neurons in the first depth level, $N_0$, needs to be defined. The depth, $D$, of the network was implemented recursively so that it could be parameterized, and was defined as the number of $max - pool$ operations and respective skip layers. The proposed changes to the $UNet$ model were implemented using Python and the TFLearn API (Damien et al. [2016]) with Tensorflow as engine (Abadi et al. [2015]).

## 4 Results and Discussion

The results were divided in two parts. In the first, a study of the $UNet$ parameters, $D$ and $N_0$, was performed. Then, the best $UNet$ model was compared with two methods present in the literature (Martins et al. [2018]), hereinafter referred as $Ref - R$ and $SM - R$.

### 4.1 Database and Metrics

A database consisting of 243 ultrasound images of the MCPJ was used (around 110 patients). The images were cropped to $(256 * 736)$ pixels to ensure no rounding errors occur in the $max - pool$ operations. The phalange, metacarpus, skin line, extensor tendon and joint capsule were manually annotated in all images, using a self owned dedicated software. It was split in 193 images to train and 50 different images for testing. The division was done manually to ensure that different cases are present in each set. The database is relatively small and, because of that, a random split method might create bias in the results. The manual division took into account the synovitis grade and the presence of other conditions such as osteophytes and bone erosions. The test set was used to compare the best $UNet$, obtained from the parameter selection, and the two reference methods. The $DICE$ will be used as a comparative metric.

### 4.2 Model Size Optimization

As previously said, the model size is an important aspect to take into account. Here we tested two main $UNet$ size related parameters, the depth, $D$ and the number of initial filter, $N_0$. From the training set, 30% of the images were selected for validation, and the results are presented in Table 1.

Table 1: Mean $DICE$ results in the validation set. In bold are the 3 best results.

| $D$ \ $N_0$ | 2 | 4 | 8 | 16 | 32 |
|---|---|---|---|---|---|
| 2 | 0.643 | 0.671 | 0.663 | 0.668 | 0.673 |
| 3 | 0.714 | 0.702 | 0.715 | 0.728 | 0.739 |
| 4 | 0.765 | 0.766 | **0.795** | 0.771 | 0.786 |
| 5 | **0.820** | **0.795** | 0.739 | 0.786 | 0.731 |

From the analysis of Table 1, it is possible to observe that the model with the higher $DICE$ was the one with $D = 5$ and $N_0 = 2$, followed by the models with $D = 5$ and $N_0 = 4$ and $D = 4$ and $N_0 = 8$. In general, the results improved as the depth increases from 2 to 4, while in the depth 5 that trend is not verified. Also, the results are more sensitive to the depth of the network than to the number of filters. Given the results presented in Table 1, one could argue that the $D = 6$ should be tested as well, but due to hardware limitations (GPU memory) this test was not performed. In Fig. 4, it is possible to visualize the outputs of the best configuration for each depth, $D$.

$$D = 2; N_0 = 32$$

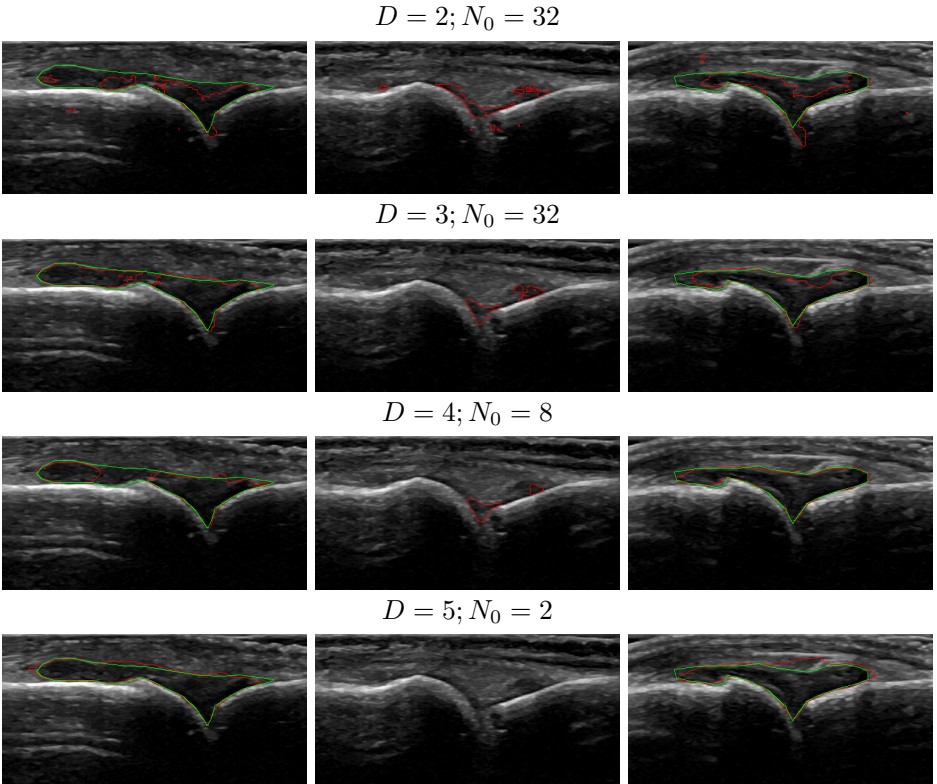

Figure 4: Visual results obtained for the segmentation of the joint capsule for different $UNet$ parameterizations (on top of the images are the values of $D$ and $N_0$). Each column presents a different case. The delineations is green and red refer to the manual and automatic segmentations, respectively.

From the analysis of Fig. 4, it is clearly visible the effect of increasing $D$; for small values the output has some granularity, while higher values of $D$ result in smoother outputs. For small values of $D$, it is also possible to observe some scattered false positives, which would be easy to remove in a post-processing step. However, the results would hardly compare to the ones of higher $D$ values.

### 4.3 Comparative Results

In this section, the best $UNet$ model will be compared with the other two existing works in the literature, $Ref - R$ and $SM - R$. In Fig. 6, it is possible to see the $DICE$'s Boxplot obtained from the 50 test images:

From Fig. 5, it is visible that the $UNet$ model achieved better results than the $SM - R$ and $Ref - R$ methods. The $UNet$ has higher $DICE$ values and less outliers (only one) than the other methods. Both, the $SM - R$ and $Ref - R$ have two outliers with $DICE$ of 0, which correspond to images without a visible joint capsule. In these cases, these methods do not have the capacity to output an empty segmentation, resulting in false positives. The statistical significance of these results was evaluated using a $Welch's\ t - test$ and observing the respective $p$-values. The results confirmed that the $UNet$ is statistically more accurate then the $Ref - R$ and $SM - R$ methods ($p$-values of $9.2E^{-9}$ and $8.4E^{-4}$, respectively).

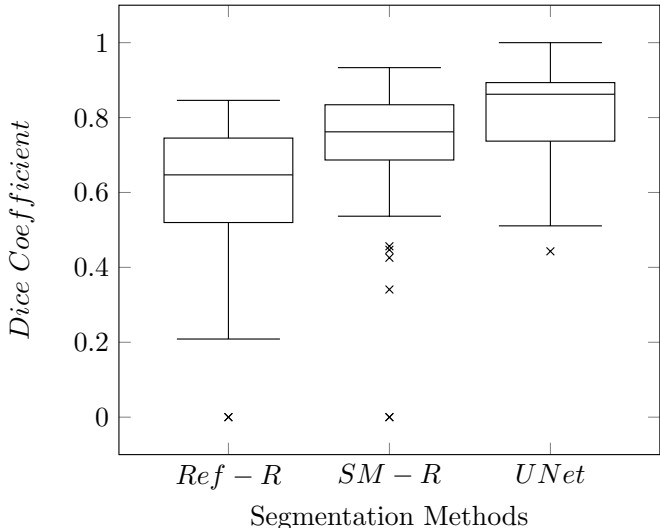

Figure 5: $DICE$'s Boxplot obtained from the test images using the $Ref - R$, $SM - R$ and $UNet$ methods. The $\times$ are the outliers.

In order to better understand the distribution of the $DICE$, it was decided to include the results of the percentage of correctly segmented images for different $DICE$ values, Fig. 6.

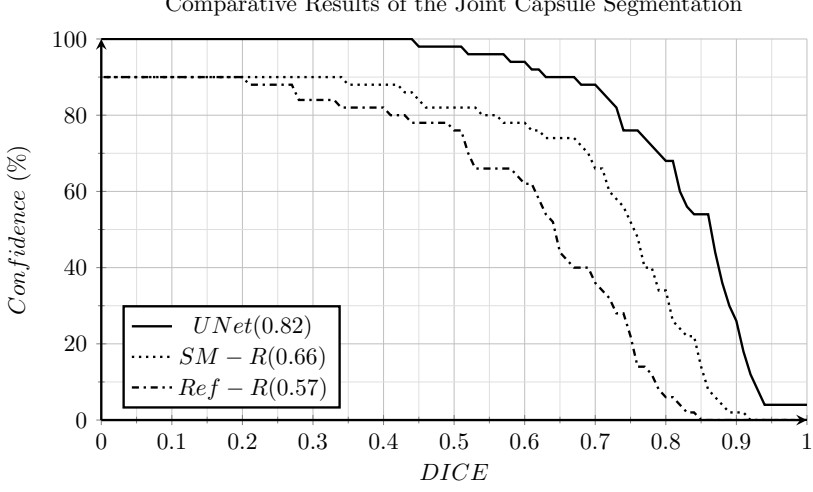

Figure 6: Segmentation results obtained for the different methods. The values between round brackets are the Area Under the Curve (AUC);

From the analysis of Fig. 6, it is possible to conclude that the $UNet$ outperforms by far the other two methods with a $Confidence$ of 90% for $DICE$s higher than 0.67, while the $Ref - R$ and the $SM - R$ achieved, for the same $Confidence$, a $DICE$ of 0.20 and 0.34, respectively. The $UNet$ model was also the only one that achieved 100% $Confidence$ for $DICE$s higher than 0.44.

For a better understanding, some visual results are presented in Fig. 7.

It is possible to verify that the $UNet$ tends to produce results that are closer to the manual annotations. The second row shows one example without joint capsule, which was correctly identified by the $UNet$. This image is responsible for the $DICE$s of 0 in the $Ref - R$ and $SM - R$ observed in Fig.

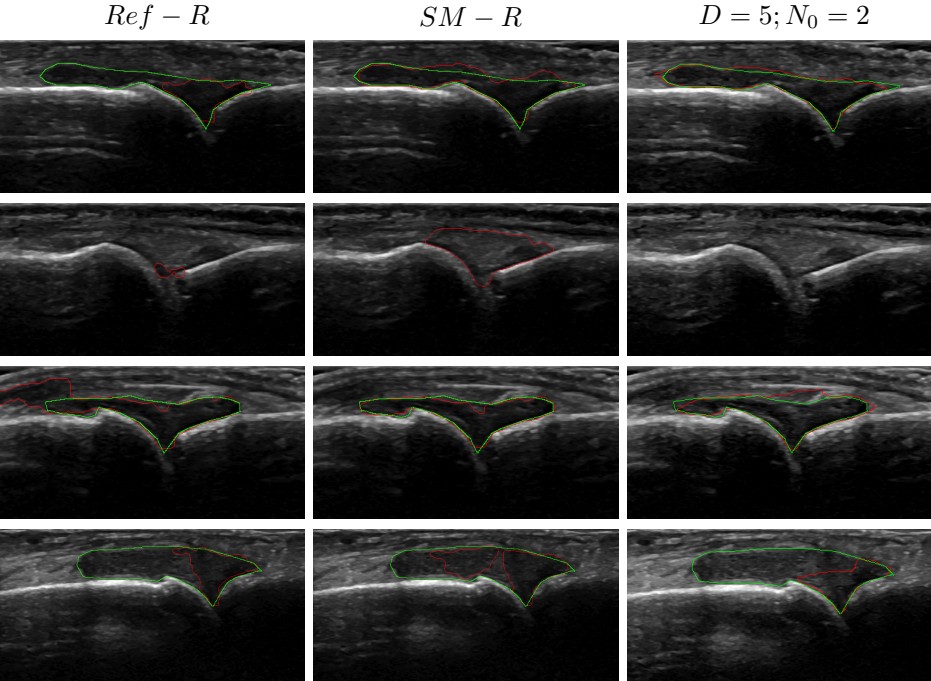

Figure 7: Visual results obtained for the segmentation of the joint capsule for the $Ref - R$, $SM - R$ and $UNet$ with $D = 5$ and $N_0 = 2$. In the columns are different cases, in green is the ground truth and in red is the respective automatic results.

5. In the fourth row, an example where the $UNet$ achieved worst results is shown (outlier of Fig. 5). In this case, the $UNet$ was not able to properly identify the joint capsule, even though it seems a trivial case, due to the reduced number of images with grade 3 synovitis and with a swelling this severe. In this example, the $SM - R$ was able to achieve better results, but they are still far from optimal.

## 5 Conclusions and Future Work

A new method for the segmentation of the joint capsule in ultrasound images of the MCPJ was proposed. Results have shown that the $UNet$ model can segment 90% of the images with a $DICE$ higher than 0.67. The $UNet$ outperforms the other two reference methods by a large margin. Furthermore, the $UNet$ model only uses the information of the image itself, in contrast to the other methods, which require additional inputs to create the region of interest.

One advantage of the $Ref - R$ and $SM - R$ methods over the $UNet$ is that the formulation and parameter selection are easier to understand and have direct implications on the results. Moreover, the $UNet$ is a black box system, which is not always well accepted by some clinical personnel. The $SM - R$ and $Ref - R$ methods assume a perfect scenario, since the manual annotations were used. Knowing that no perfect automatic segmentation method exists for these structures, it is expected that the results of these methods deteriorate once the automatic identifications are added.

In the future, the database will be expanded with new examples, with a focus in images with pathology. Additional data augmentation techniques may also be explored, as well as transfer learning. The inclusion of the segmentation results in an automatic synovitis detection and quantification system will also be done.

**Acknowledgments**

Work supported by European Union funds through the FUNDO EUROPEU DE DESENVOLVI-MENTO REGIONAL and by national funds by Quadro de Referência Estratégico Nacional in the scope of Project Rheumus (Projeto QREN no: 38505). This work also has contributions from the

project NanoSTIMA, NORTE-01-0145-FEDER-000016, supported by Norte Portugal Regional Operational Program (NORTE 2020), through Portugal 2020 and the European Regional Development Fund (ERDF).

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
