# OpenReview forum: "Joint Capsule Segmentation in Ultrasound Images of the Metacarpophalangeal Joint using Convolutional Neural Networks"
_MIDL.amsterdam/2018/Conference — Submitted to MIDL 2018_

### Review · AnonReviewer2 · 2018-05-08
**Review of Joint Capsule Segmentation in Ultrasound Images of the Metacarpophalangeal Joint using Convolutional Neural Networks**

**Rating:** 2
**Confidence:** 3

**Review:**

The authors propose an automatic segmentation of the joint capsule in ultrasound images of the metacarpophalangeal joint using an adapted version of the well known UNet model. This work compares several previous methods (Ref-R, SM-R) for segmentation. Experimental results are relatively extensive, but they lack novelty.
1. Please compare your method with original Unet and/or densenet with various kinds of evaluation metrics including JACCARD, MSD, time, etc.
2. It is seldom new contributions. The adapted UNet is to chain zero padding, location of normalization layer, sigmoid from softmax and add regularization.
3. This results look over-fitted to this dataset. Extra-validation is needed.


**Special Issue:**

No

---

### Review · AnonReviewer1 · 2018-05-10
**U-Nets for a 2D ultrasound segmentation task**

**Rating:** 1
**Confidence:** 2

**Review:**

The authors discuss and evaluate the use of the U-Net architecture for the segmentation of joint capsules in 2D ultrasound images. The paper is well written and easy to read.
Two parameters of the U-Net (the depth of the network and the number of kernels) are optimized on a validation set and results of this optimization are reported. The best performing network is compared on an independent test set against two existing methods. The U-Net outperforms the existing methods, but it must be said that the two existing methods are not based on state-of-the-art image analysis methodology. The existing methods use a combination of region growing and split and merge techniques.
The quantitative and qualitative results of the U-net method are not impressive and not what you would expect from a modern deep learning based method. The top 90% of the results have a DICE score of only 67% or higher. Also, the images shown in the paper show the method not performing so well qualitatively. The U-net segmentation seem to be much more noisy than the manual segmentations and it also seems to sometimes miss structures that are relatively clear (e.g. the bottom right image in figure 5). It’s expected that with more modern deep learning techniques a better result could have been obtained.
The educational value in this paper can be summarized as a confirmation that U-nets work well and that for this application (with not so much training data) a not extremely shallow network with a relative low amount of parameters works well.


**Special Issue:**

No

---

### Review · AnonReviewer3 · 2018-05-11
**Using U-net for segmentation of metacarpophalangeal joint**

**Rating:** 1
**Confidence:** 2

**Review:**

The authors use an adjusted version of U-net to segment the joint capsule in 2D ultrasound images. The authors introduce five augmentations to the compensate for the reduced number of training samples. The authors also made three adjustments to the U-net introduced by Ronneberger et al.
- Invalid padding to simplify the model
- Introduce batch normalization
- Replace the Softmax with a Sigmoid function.
Next to this, to authors evaluated two parameters of the U-net: the number of initial filters and the depth of the network.

Major issues:
- The paper does not include any methodological novelties and although the method is compared to two other methods (REF - R and SM - R), the results are not very impressive.
- No results are shown that evaluate the five introduced augmentations, so it is not clear if these augmentations are valid and that they improve system performance.
- The authors conclude from the results in Table 1 (obtained from the validation set of ~58 images) that network with a depth of 5 and 2 initial filters gives the best performance. The results in Figure 4 seem to indicate that the receptive field below a depth of 5 is insufficient for this task. The authors state that “one could argue that the D=6 should be tested as well, but due to hardware limitations this test was not performed”. There are other ways to increase the receptive field of the U-net (e.g. downsampling the input image or using dilated convolutions) which were not evaluated in this paper.

Minor issues:
- The authors compare their method to two other methods (Ref – R and SM – R), but the sentence that introduces these two abbreviations does not clearly describe which methods they refer to (there is only one citation of the work of Martins et al.).
- There are multiple images per patient, but it is not stated if these test images were separated on patient level.
- The Welch’s t-test is used for statistical comparison, but it is not explained why this method is chosen instead of the student-t test (was the variance unequal?).
- The y-axis of Figure 6 presents the ‘confidence’, but the text does not clearly describe the meaning of this parameter.


**Special Issue:**

No

---

### Decision · Program_Chairs · 2018-05-15
**Paper33 Acceptance Decision**

Reject